# Recent Progress in Dimerized Small-Molecular Acceptors for Organic Solar Cells

**DOI:** 10.3390/molecules30071630

**Published:** 2025-04-06

**Authors:** Xin Tang, Yamin Zhang, Hao-Li Zhang

**Affiliations:** State Key Laboratory of Applied Organic Chemistry (SKLAOC), Key Laboratory of Special Function Materials and Structure Design (MOE), College of Chemistry and Chemical Engineering, Lanzhou University, Lanzhou 730000, China; txin2023@lzu.edu.cn

**Keywords:** organic solar cells, dimerized small-molecular acceptor, stability, efficiency

## Abstract

Organic solar cells (OSCs) have witnessed significant advancements in recent years, largely propelled by innovations in material design and device engineering. Among the emerging materials, dimerized small-molecule acceptors (DSMAs) have garnered considerable attention due to their unique advantages. For instance, DSMAs can directly inherit the excellent optoelectronic properties of corresponding small-molecule monomers. Moreover, their relatively larger molecular weight can effectively suppress molecular diffusion in the active layer, thereby enhancing the stability of OSCs. Compared to polymer acceptors, DSMAs have a well-defined structure, which is free from batch-to-batch variability, greatly enhancing the reproducibility of devices. This review comprehensively summarizes recent progress in DSMAs for OSCs, with a focus on their two primary linkage configurations: conjugative and non-conjugative connections. Additionally, the impact of various connection positions (including core-unit, end-group, and side-chain connection sites) on molecular packing, optoelectronic properties, and device performance is systematically reviewed. The review highlights the critical role of DSMAs in addressing key challenges in OSCs, such as photodegradation and morphological instability, while balancing power conversion efficiency and long-term stability. By consolidating recent breakthroughs and identifying future research directions, this work aims to provide valuable insights into the rational design of DSMAs, paving the way for the development of high-performance and commercially viable OSCs.

## 1. Introduction

The emergence of photovoltaic (PV) technology has provided a promising pathway to address the energy crisis and environmental challenges [1,2,3]. Among various photovoltaic technologies, organic solar cells (OSCs) have attracted considerable attention due to their unique advantages, including flexibility, lightweight design, solution processability, and potential for large-scale fabrication [4,5,6]. Unlike silicon solar cells, organic solar cells generate electron-hole pairs (known as excitons) rather than free electrons when the photoactive layer materials absorb photons. Due to the low dielectric constant of organic materials, these excitons need to be dissociated at the interface between two materials with different electron affinities: an electron donor and an electron acceptor. The generated holes and electrons then move towards the corresponding interfacial layers, respectively, for further transport to the electrodes [7]. The enhancement of power conversion efficiency (PCE) in organic photovoltaic (OPV) devices is contingent upon the synergistic progress in photoactive layer materials and device engineering strategies [8,9,10]. Acceptor materials have undergone significant evolution, transitioning from fullerene-based derivatives to electron acceptors based on aromatic diimide, and subsequently to small-molecule acceptors (SMAs) [11,12]. Over the past decades, significant progress has been made in the development of SMAs, particularly those with an acceptor–donor–acceptor (A–D–A) structure, such as ITIC and Y6, which have become a focal point in the pursuit of high-performance OSCs [13,14,15,16]. Although state-of-the-art OSCs based on fused-ring SMAs have achieved remarkable power conversion efficiencies (PCEs) surpassing 20%, challenges related to stability remain critical barriers to their large-scale industrial application [17,18]. The stability of organic photovoltaic devices is primarily governed by the properties of the active layer, with chemical degradation of materials and their migration within the active layer being the main factors contributing to performance decline in OSCs. External environmental stressors, such as light exposure, thermal effects, and mechanical strain, further exacerbate these degradation processes [19,20,21].

Currently, the combination of polymer donors and small-molecule acceptors in the active layer of OSCs represents the most efficient configuration. However, SMA-based OSCs often suffer from significant morphological degradation, largely due to the small molecular size, rapid diffusion kinetics, and low glass transition temperature (Tg) of SMAs, which collectively result in significant thermal-driven migration and aggregation in the blend phase, leading to undesirable phase separation and performance degradation over time [22,23]. To address these limitations, in 2021, Zhang and Li et al. developed the first polymerized small-molecule acceptors (PSMAs) to simultaneously enhance device efficiency and morphological stability [24]. However, PSMAs face significant challenges, including batch-to-batch variability and reproducibility issues, which hinder their practical application in large-scale production [25,26,27,28,29,30].

To obtain acceptor materials that combine the advantages of SMAs and PSMAs, dimerized small-molecular acceptors (DSMAs) have emerged as a promising alternative [31]. DSMAs exhibit well-defined molecular structures, excellent batch reproducibility, superior film-forming properties, low diffusion rates, and enhanced stability, making them a highly attractive material platform for achieving high-performance and stable OSCs. Herein, we comprehensively review recent advancements in DSMAs for OSCs, with a focused analysis of their two fundamental linkage configurations: conjugative and non-conjugative connections. We systematically analyze the structural implications of diverse connection topologies—including core–core, end-group–end-group, and side-chain–side-chain dimeric configurations—on molecular packing behavior, optoelectronic characteristics, and photovoltaic device performance. This critical evaluation elucidates essential design principles for optimizing DSMA systems. Moreover, the review underscores the transformative potential of DSMAs in mitigating persistent challenges in OSC development, notably photochemical degradation and thermodynamic instability in bulk heterojunction morphologies.

## 2. Conjugatively Linked DSMAs

Rigid backbones have been demonstrated to enhance electronic conduction by providing well-defined pathways for charge carriers. Therefore, rigid conjugated extensions facilitate intramolecular charge transfer and the formation of electron transport networks. Conjugatively linked DSMAs can effectively modulate molecular aggregation behavior through variations in linkage group and positions, enabling the construction of efficient charge transport networks and the formation of optimal fibrillar interpenetrating network structures with the donor materials. Figure 1 illustrates the structures of representative conjugatively linked DSMA molecules. The photovoltaic parameters of the conjugatively linked DSMAs are shown in Table 1.

### 2.1. Conjugatively Linked DSMAs Based on Fused-Ring Strategy

The ring-fused strategy for the design of DSMAs can effectively extend the conjugated system, thereby improving electron transport and reducing non-radiative energy losses. In 2022, Chen et al. synthesized a two-dimensional A–D–A-structured acceptor, named CH8, featuring four electron-withdrawing terminal groups. The expanded conjugation in both dimensions endows CH8 with an exceptionally low electron reorganization energy of 98 meV. This characteristic positions CH8 as a promising candidate for use in high-performance organic semiconductor materials. The device, based on PM6: CH8, achieved a considerate PCE of 9.37% with a high open-circuit voltage (*V*_oc_) of 0.889 V and low energy loss (Eloss) below 0.6 eV [32].

In 2024, Li et al. developed a pyrene-fused dimerized electron acceptor (DP-BTP, as shown in Figure 1) that deviates from the conventional linearly configured dimerized acceptors. This unique acceptor boasts a distinctive “butterflylike” architecture, with two Y-shaped acceptors acting as wings connected through a pyrene-based central structure. This design allows the new dimerized acceptor to be seamlessly integrated as the third component in ternary OSCs, significantly enhancing electron transport capabilities, minimizing non-radiative voltage losses, and increasing the open-circuit voltage. These advancements have led to a remarkable efficiency of 19.07% in ternary OSCs, a substantial increase from the 17.6% efficiency observed in binary OSCs. Most notably, the high Tg of the pyrene-fused electron acceptor contributes to the stabilization of the photoactive layer’s morphology, even after thermal treatment at 70 °C, maintaining an impressive 88.7% of its initial efficiency over a 600-h period [33]. These results indicate that DSMAs with conjugated connections via the fused-ring strategy are capable of enhancing device performance while maintaining high thermal stability.

### 2.2. Conjugatively Linked DSMAs Based on Non-Fused-Ring Strategy

DSMAs with conjugated connections via the fused-ring strategy have demonstrated potential for enhancing device thermal stability. However, their synthesis is challenging, and the substrates are relatively limited. Therefore, researchers have developed a series of DSMAs with conjugated connections via a non-fused-ring strategy. This approach not only allows for the conjugation extension of the acceptor molecules but also offers a variety of linking sites (as shown in Figure 1), which can further modulate the packing mode and optoelectronic properties of DSMAs.

The first category of conjugatively linked DSMAs based on non-fused-ring strategy is similar to DSMAs with conjugated connections via fused-ring strategy, both of which achieve conjugation extension through the core unit. In 2023, Chen et al. synthesized a series of multi-dimensional non-fullerene acceptors with extensive conjugated extensions in various directions, namely CH8-0, CH8-1, and CH8-2. These 3D NFAs, featuring specific A–D–A architectures in two distinct directions, exhibit exceptionally low reorganization energy, a fibrillar network film morphology, enhanced charge transport properties, and improved stability. By employing molecular geometry control through fluorine-induced noncovalent conformational locks, the CH8-1-based binary bulk heterojunction OSCs have achieved an impressive power conversion efficiency of 17.05% [34].

The second category of conjugatively linked DSMAs based on non-fused-ring strategy is linked through the β-side chain. In 2024, Chen et al. synthesized a novel dimerized small-molecule acceptor named BC-Th, which is distinctive for its branch coupling (BC-type) approach, where the branched groups of two monomers are linked, diverging from the standard terminal coupling (TC-type) method. This innovative design endows BC-Th with a three-dimensional structure and less restricted terminal groups, resulting in diverse molecular orientations, an increased dielectric constant, and enhanced electron mobility. Consequently, the ternary OSC incorporating D18: L8-BO: BC-Th has achieved a remarkable PCE of 17.43% and an impressive FF of 80.11% [35].

To delve deeper into the impact of conjugation length and its modifications on molecular properties, they synthesized two branch-connected dimerized acceptors, designated as D1 and D2. D1 utilizes bithiophene as a linker group, while D2 employs difluorinated bithiophene for the same purpose. Notably, D2 exhibits a higher molar extinction coefficient and, more significantly, superior nanoscale film morphology and enhanced charge transport properties compared to D1. Consequently, the device based on D2 achieved an impressive PCE of 16.66%. This study underscores the critical role of linking group selection in the development of high-performance, branch-connected dimerized acceptors [36].

In the construction of conjugatively linked DSMAs using the non-fused-ring strategy, the end-group linkage method is the most frequently employed approach. In 2022, He et al. synthesized a novel dimerized small-molecule acceptor, dBTICγ-EH, by oligomerizing small molecules. This dimer, dBTICγ-EH, demonstrated remarkable performance in bulk heterojunction devices, achieving PCEs of 14.48%. Moreover, it exhibited an impressive T80% lifetime of 1020 h under continuous illumination. These results significantly outperform those of both small molecular and polymeric acceptors, highlighting the potential of dBTICγ-EH in the field of OSCs [31]. They further developed a thiophene terminal-linked DSMA, DTY-2Cl, which boasts high planarity due to its bithiophene linkage. Theoretical calculations and single-crystal analysis of the regional configuration have revealed that the twisting angle between the bithiophene binding units in DTY-2Cl is nearly negligible, being almost zero. Consequently, devices fabricated with DTY-2Cl in combination with D18 exhibited exceptional photoelectric performance, achieving a PCE of 18.06%. In addition to this high efficiency, the devices have also demonstrated excellent stability, further underscoring the potential of DTY-2Cl in the advancement of organic photovoltaic technology [37].

Zhou et al. introduced an innovative strategy for designing electron acceptors and have developed a “quasi-macromolecule” (QM) featuring an A–π–A architecture. In this design, a functionalized π-bridge serves as a connector between two small molecular acceptors, denoted as A, enhancing long-term stability without compromising device efficiency. This A–π–A-type QM offers remarkable synthetic versatility, allowing for the fine-tuning of optical and electrochemical properties, as well as crystallization and aggregation behaviors, by varying the A and π components. Devices fabricated with the synthesized A–π–A-type acceptor, QM1, demonstrate superior stability and impressive power conversion efficiencies of up to 17.05%. The “molecular-framework” A–π–A structure presents a novel approach to the development of photovoltaic materials that are both highly efficient and stable [38] (as shown in Figure 2).

To further investigate the effect of conjugation length on molecular morphology and device performance, the researchers synthesized three distinct acceptors—QM-1T, QM-2T, and QM-3T—each featuring a π-bridge with varying degrees of conjugation length, comprising thiophene, bithiophene, and terthiophene, respectively. They meticulously investigated how the molecular size, influenced by the π-bridge units, could be finely controlled to affect the morphology of the active layer and, consequently, the performance of the devices. The blend of PM6 with QM-2T exhibited an exceptionally well-controlled morphology, which led to significantly enhanced and balanced electron and hole mobilities. As a result, the OSCs based on QM-2T achieved a high open-circuit voltage of 0.94 V, without a reduction in short-circuit current density, culminating in an improved device efficiency of 17.86% [39].

Wei et al. designed an N–π–N-type oligomeric acceptor 2BTP-2F-T, constructed by two small non-fullerene acceptor units linked with a thiophene π bridge. The 2BTP-2F-T compound not only harnesses the benefits of both small-molecule non-fullerene acceptors and polymeric acceptors (PYF-T-o) that share similar units, but it also showcases superior attributes such as a high absorption coefficient and elevated electron mobility (μ_e_), with reduced reliance on molecular packing. When paired with PM6 as the donor material, an impressive efficiency of 18.19% is achieved, featuring an open-circuit voltage of 0.911 V, a short-circuit current density (*J*_sc_) of 25.50 mA cm^−2^, and a fill factor (FF) of 78.3%. These performance metrics significantly outperform those of the corresponding monomer-based devices (16.54%) and PYF-T-o-based devices (15.8%) [40].

To explore the molecular conformation of DSMA and its influence on molecular stacking, Wang et al. synthesized a dimerized acceptor material, DIBP3F-Se, which connects two segments of Y6-derivatives via selenophene. Through theoretical simulations and experimental 1D and 2D NMR spectroscopic studies, they demonstrated that the dimer adopts an O-shaped conformation, distinct from the S- or U-shaped conformations typically observed. This unique O-shaped conformation is believed to be controlled by a specific “conformational lock” mechanism, which is attributed to the enhanced intramolecular π–π interactions between the two terminal groups within the dimer. The OSC incorporating DIBP3F-Se achieved a PCE of 18.09% [41,48]. Ge et al. synthesized a novel giant-molecule acceptor (GMA), designated as GMA-SSeS, by meticulously selecting the number and placement of selenium atoms, π-spacer linking units, and the outermost conjugate ring of the central core in monomers. The OSC based on PM6: GMA-SSeS achieved a remarkable PCE of 19.37%, with an impressive *V*_oc_ of 0.917 V, which corresponds to a reduced voltage loss (Δ*E*_3_) of 0.246 eV, and an excellent FF of 77.12%. In addition to these performance metrics, the GMA-SSeS molecule demonstrated excellent thermal stability, with an impressively extended T_80_ lifetime of 5600 h. This study underscores the potential of selenium substitution in GMA structures, both in linking units and monomers, as a valuable strategy for developing high-performance and long-lasting stability devices [42].

The photovoltaic properties and stabilities of DSMA materials could be affected by the linking groups used. Kim et al. synthesized a dimerized small-molecule acceptor, DYBO, by employing the electron-donating BDT linker to connect two SMA units derived from Y-analogues. The device incorporating PM6: DYBO demonstrated enhanced operational stability under thermal stress and illumination. This can be attributed to two factors: (1) the extended molecular length of DYBO, which inhibits molecular diffusion and consequently raises the Tg; and (2) the BDT linker, which is also a component of PM6, facilitating improved molecular interactions with the polymer donor [43]. Li et al. developed two distinct vinyl π-spacer linking-site isomerized giant-molecule acceptors, with the π-spacer positioned on either the inner carbon (EV-i) or the outer carbon (EV-o) of the benzene end group in the SMA, featuring longer alkyl side chains (ECOD) to enable processing with non-halogenated solvents. Notably, EV-i exhibited a twisted molecular structure that paradoxically enhances conjugation. OSCs utilizing EV-i as the acceptor and processed with the non-halogenated solvent o-xylene have achieved an impressive PCE of 18.27% [44].

Sun et al. developed a pair of isomeric dimer acceptors, D-TPh and its counterpart, which are differentiated by the spatial orientation of their end capping groups. The D-TPh isomer stands out for its improved backbone planarity, a higher lowest unoccupied molecular orbital (LUMO) energy level, and a more orderly molecular stacking arrangement. These structural features have led to remarkable performance in OSC devices, with the PM6: D-TPh blend achieving a PCE of 19.05%. Furthermore, the PM6:D-TPh device has shown exceptional stability, with an extrapolated T80 lifetime exceeding 2800 h under continuous one-sun illumination and maximum power point (MPP) tracking [45]. In contrast to the DMSA linked through the donor unit, Zhang et al. delved into the realm of n-type linkers by utilizing the benzothiadiazole unit to dimerize the SMA units through a click-like Knoevenagel condensation reaction, resulting in the formation of BT-DL. This novel dimer, BT-DL, demonstrates a markedly enhanced intramolecular super-exchange coupling, a highly beneficial characteristic for the acceptor component in organic photovoltaics. It has shown remarkable performance, achieving a high PCE of 18.49%. Most notably, devices based on BT-DL display extraordinary stability, maintaining over 90% of their initial efficiency after being subjected to 1000 h of thermal stress at 90 °C [46]. In 2025, Bao et al. synthesized two DSMA, DY-DPP and DY-IDG, utilizing diketopyrrolopyrrole (DPP) and isoindigo (IDG) derivatives as fundamental electron-withdrawing building blocks. Notably, the bulk heterojunction device incorporating DY-IDG as the acceptor material with PM6 donor polymer achieved an impressive power conversion efficiency of 14.01%, underscoring the potential of IDG-based molecular engineering in high-performance organic photovoltaics [47].

## 3. Non-Conjugatively Linked DSMAs

In addition to using the conjugation extension method, incorporating flexible linkers to construct non-conjugatively linked DSMAs is also an important design strategy for DSMAs. DSMAs with different flexible spacers have been reported to regulate their aggregation and relaxation behavior. Additionally, the flexible spacer can entangle with polymer donor molecules, thereby improving the photothermal stability of the active layer. Moreover, DSMAs based on flexible spacers tend to form early aggregation in the blended film, which effectively counteracts efficiency degradation caused by molecular diffusion. Figure 3 illustrates the structures of representative non-conjugatively linked DSMA molecules. The photovoltaic parameters of the non-conjugatively linked DSMAs are shown in Table 2.

The first category of non-conjugatively linked DSMAs involves connecting two SMAs through the core unit via flexible spacers. Li et al. pioneered the synthesis of the first dimerized non-fused electron acceptor, utilizing Thieno[3,4-c]pyrrole-4,6-dione as the central core. Despite having similar energy levels, the dimerized acceptor (RM2) and its monomeric counterpart display distinct absorption spectra, a result of their differing aggregation behaviors. OSC incorporating the dimerized acceptor have shown improved performance, with a high PCE of 11.05%, and have also demonstrated enhanced thermal stability [49].

Min et al. synthesized a novel dumbbell-shaped dimerized acceptor, DT19, featuring a hinge-like structure that enables it to form a semi-alloy with the host acceptor. This unique configuration results in robust interchain entanglement with the polymer donor, effectively mitigating phase separation and excessive aggregation under thermal stress. As a result, the PM1:BTP-eC9 system based on DT19 demonstrated remarkable stability, maintaining over 90% of its initial efficiency after being subjected to 120 °C for 200 h [50].

Chen et al. synthesized three back-to-back connected dimers—2Qx-TT, 2Qx-C3, and 2Qx-C6—to serve as guest acceptors in organic solar cells, enhancing their overall performance. By modulating the linkage from a rigid bithiophene to a flexible alkyl chain, these dimers exhibit significantly different molecular geometries and intermolecular interactions. These differences subsequently affect their packing arrangements, film-forming processes, carrier mobilities, and the efficiency, stability, and flexibility of the devices. Among them, the 2Qx-C3-based ternary device achieved the highest efficiency, reaching an impressive PCE of 19.03% [51]. He et al. chose thienylenealkane-thienylene (TAT) as the conjugate-break linker and synthesized four dimerized acceptors by strategically varying the connecting sites and halogen substitutions. This molecular structure optimization led to a rational phase separation within the blend films for the dimerized acceptor. This phase separation significantly promoted exciton dissociation while effectively mitigating charge recombination processes. As a result, the FDY-m-TAT-based OSC achieved a remarkable PCE of 18.07% [52].

In 2023, Zhang et al. proposed the concept of tethered small-molecule acceptors (TSMAs) by covalently tethering SMAs via the pyrrole unit of Y-series SMAs. This represents an environmentally friendly and effective synthesis strategy for dimeric acceptors. They introduced a series of DSMAs named DY-1, DY-2, and DY-3, each engineered with different flexible spacers to modulate their aggregation and relaxation dynamics. A key discovery is that the SMAs, when tethered, exhibit significantly higher glass transition temperatures, which effectively suppress thermodynamic relaxation within mixed domains. The dimer-based devices, particularly those based on DY-2, have shown a substantial reduction in burn-in efficiency loss, with the DY-2-based device retaining over 80% of its initial PCE after 700 h of operation at the maximum power point (MPP) under continuous illumination. These results indicate that the tethered Y6-based blend forms a less miscible system compared to the PM6: Y6 blend, with the diffusion-enabled demixing of the morphology being significantly suppressed. This leads to a device that is thermodynamically more stable, offering promising implications for the longevity and reliability of organic photovoltaic devices [53]. Subsequently, they synthesized the DY-P2EH molecule by incorporating phenyl groups into the alkyl side chains. The ternary device based on PM6: BTP-ec9: DY-P2EH achieves a remarkable FF of 80.61% and an impressive PCE of 19.09%. Moreover, the ternary device demonstrated exceptional stability, retaining more than 85% of its initial efficiency after being subjected to thermal stress at 85 °C for 1100 h [54].

To gain a deeper understanding of how molecular geometries affect the properties of TSMAs, Zhang et al. further regulated the molecular geometries through thiophene-core isomerism engineering and synthesized two isomeric DSMAs, namely TDY-α and TDY-β. These tethered dimers exhibited folded geometries in solution, with the overlapping preference of individual Y6 subunits being influenced by the bent shape of the aromatic core. This architectural feature results in distinct electronic coupling, thermodynamic properties, and aggregation behaviors for the two dimers, as shown in Figure 4. TDY-α, in particular, demonstrated a higher glass transition temperature, improved crystallinity, and an optimal Flory–Huggins interaction parameter with the polymer donor. Consequently, devices based on TDY-α showed a significant increase in efficiency, reaching up to 18.1%. Crucially, TDY-α-based devices experienced a substantially reduced burn-in efficiency loss compared to Y6 and its isomer TDY-β. They retained over 80% of their initial power conversion efficiency under long-term annealing at 80 °C for 1000 h or under continuous illumination for 1100 h [55].

To investigate the impact of different linking sites on the performance of DSMAs, in 2024, Fan et al. developed three non-fully conjugated dimerized giant acceptors, (named 2Y-sites, including wing-site-linked (linked through the β-side chain) 2Y-wing, core-site-linked 2Y-core, and end-site-linked 2Y-end). Among these, the 2Y-wing variant demonstrated enhanced miscibility, which in turn optimized the morphology and facilitated more efficient charge transfer. Consequently, OSCs based on D18/2Y-wing achieved an impressive PCE of 17.73%. Building on the success of 2Y-wing in binary systems, the author extended its application to ternary OSCs by adding it to the D18/BS3TSe-4F host system; as a result, the ternary OSCs obtain a higher PCE of 19.13 %, setting a new efficiency benchmark for the dimer-derived OSCs [56].

## 4. Conclusions and Outlook

Despite the relatively short research history of dimerized small-molecular acceptors, this molecular design strategy has rapidly demonstrated significant advantages since its emergence. This review systematically summarizes the current design strategies for DSMAs, focusing on the perspectives of linking modes and positions. The two predominant linking strategies have distinct merits: Conjugatively linked DSMAs allow for precise modulation of aggregation behavior and charge transport networks by tailoring linkage positions and functional groups, which facilitates the formation of efficient fibrillar interpenetrating networks with donors. Exemplifying this approach, the dimeric acceptor GMA-SSeS, featuring end-group linkage, achieved a binary device efficiency of 19.37% with a T80 lifetime of 5600 h. In contrast, non-conjugatively linked DSMAs utilize flexible spacers to control molecular aggregation and diffusion dynamics. This approach enhances photothermal stability, mechanical flexibility, and compatibility with polymeric donors, thereby improving fill factors. The pyrrole-linked dimeric acceptor (tethered small-molecule acceptors) TDY-α demonstrated a binary device efficiency of 18.10% and an extrapolated T80 lifetime of ~35,000 h (~15 years) under real-world outdoor conditions in Beijing (7 h/day operation). These results underscore the potential of DSMAs in advancing the efficiency and stability of organic photovoltaics.

However, most DSMAs are derived from Y-series small-molecule acceptors, which lead to increased structural complexity, higher synthesis costs, and prolonged development cycles. Future efforts should prioritize the development of new monomeric acceptors with simplified structures and enhanced performance. Moreover, although the current linking groups primarily serve structural roles, the introduction of multifunctional moieties could potentially further enhance the performance of DSMAs. At the same time, although the current DSMAs have improved the morphological stability of devices, ensuring that the performance of DSMAs surpasses that of the corresponding monomeric acceptors through structural design may require reliance on artificial-intelligence-assisted molecular design methodology in the future.

In conclusion, dimerization strategies have emerged as a highly promising approach in the field of organic solar cells, offering a favorable balance between efficiency and stability. By focusing on the refinement of monomeric acceptor structures, the design of linking groups, and the optimization of connection methods, DSMAs are poised to play a pivotal role in driving the field of organic solar cells towards unprecedented levels of performance and reliability.

## Figures and Tables

**Figure 1 molecules-30-01630-f001:**
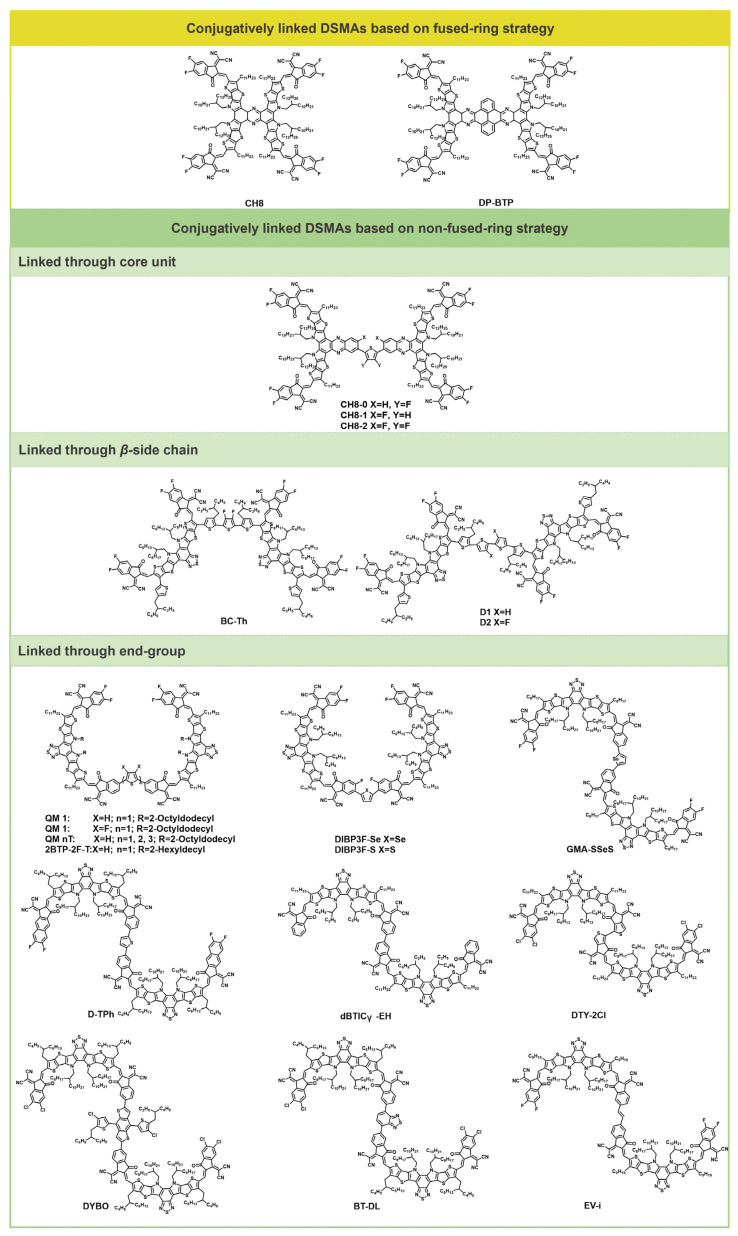
Molecular structures of conjugatively linked DSMAs.

**Figure 2 molecules-30-01630-f002:**
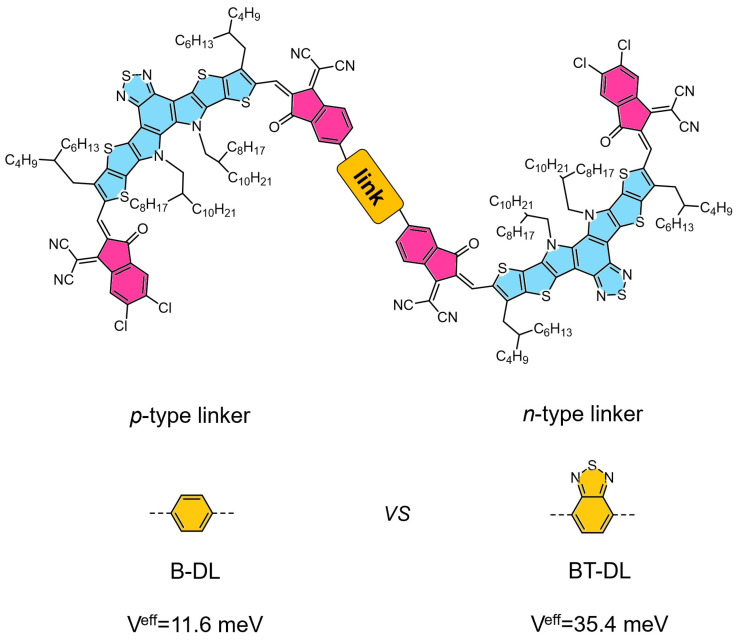
The scheme of conjugatively linked DSMAs based on non-fused-ring strategy that linked through the end groups. Reproduced with permission from ref. [46]. Copyright 2024, John Wiley and Sons.

**Figure 3 molecules-30-01630-f003:**
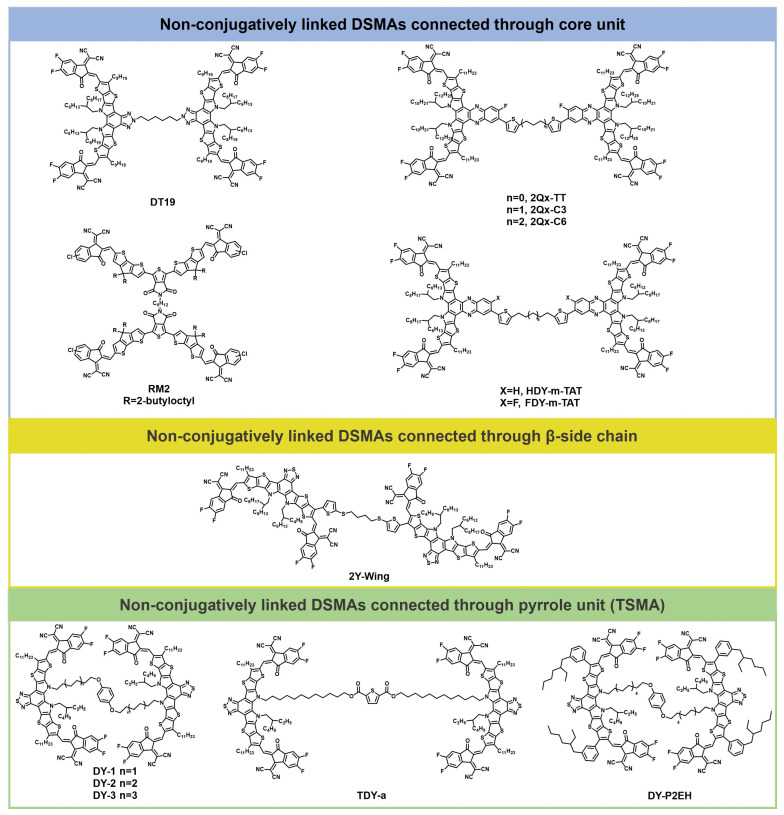
Molecular structures of non-conjugatively linked DSMAs.

**Figure 4 molecules-30-01630-f004:**
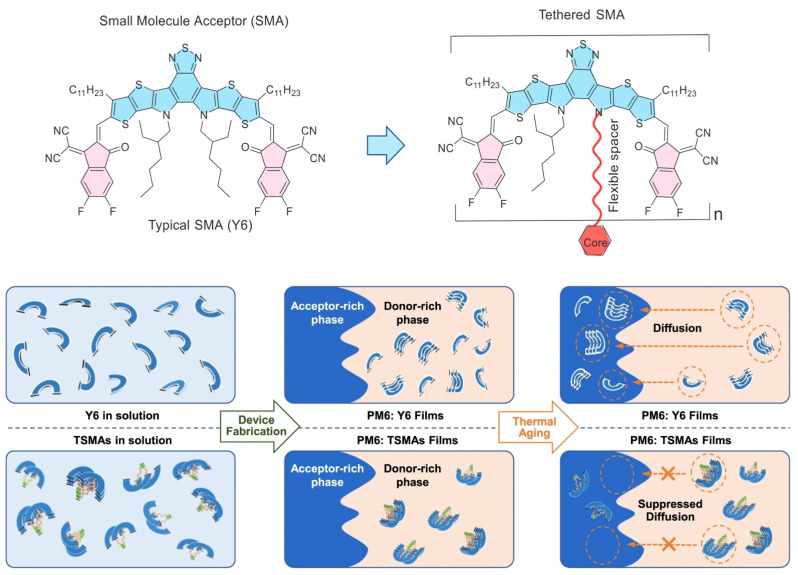
Schematic illustration of the DSMA connected via non-conjugated linkages through pyrrole unit (also named TSMAs) and Y6 molecular distribution and diffusion process in solution and PM6: acceptor films during the device fabrication and thermal aged procedures. Reproduced with permission from ref. [55]. Copyright 2023, Springer Nature.

**Table 1 molecules-30-01630-t001:** Summary of photovoltaic parameters for OSCs based on conjugatively linked DSMAs.

Active Layer	*V*_oc_ [V]	*J*_sc_ [mA cm^−2^]	FF [%]	PCE [%]	Ref.
PM6: CH8	0.889	19.70	53.50	9.37	[32]
D18: DP-BTP	0.960	22.73	69.10	15.08	[33]
PM6: CH8-1	0.923	24.89	74.20	17.05	[34]
D18: BC-Th	0.893	24.67	79.13	17.43	[35]
PM6: D2	0.876	24.33	78.17	16.66	[36]
PBQx-H-TF: dBTICγ-EH	0.910	23.41	75.49	16.06	[31]
D18: DTY-2Cl	0.913	26.2	75.46	18.06	[37]
PM6: QM1	0.910	25.23	74.01	17.05	[38]
PM6: QM-2T	0.940	25.75	73.80	17.86	[39]
PM6: 2BTP-2F-T	0.911	25.50	78.28	18.19	[40]
PM6: DIBP3F-Se	0.917	25.92	76.10	18.09	[41]
PM6: GMA-SSeS	0.917	27.38	77.12	19.37	[42]
PM6: DYBO	0.968	24.62	75.80	18.08	[43]
PM6: EV-I	0.897	26.60	76.56	18.27	[44]
PM6: D-TPh	0.946	25.59	78.7	19.05	[45]
PM6: BT-DL	0.94	25.523	77.08	18.49	[46]
DY-IDG	0.909	21.78	70.78	14.01	[47]

**Table 2 molecules-30-01630-t002:** Summary of photovoltaic parameters for OSCs based on non-conjugatively connected DSMAs.

Active Layer	*V*_oc_ [V]	*J*_sc_ [mA cm^−2^]	FF [%]	PCE [%]	Ref.
PM6: RM2	0.856	18.33	70.00	11.05	[49]
PM1: L8-BO: DT19	0.899	27.80	73.80	18.40	[50]
PM6: L8-BO: 2Qx-C3	0.898	26.45	80.10	19.03	[51]
PM6: FDY-m-TAT	0.911	26.47	74.70	18.07	[52]
PM6: DY2	0.870	26.60	76.85	17.85	[53]
PM6: DY-P2EH	0.905	24.03	78.58	17.09	[54]
PM6: TDY-α	0.864	26.90	78.00	18.10	[55]
D18: 2Y-wing	0.850	27.66	75.40	17.73	[56]

## Data Availability

The data that support the findings of this study are available from the corresponding author under reasonable request.

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
