# Peer review of "Recent Progress in Dimerized Small-Molecular Acceptors for Organic Solar Cells"

_molecules, 2025, doi:10.3390/molecules30071630_

Round 1
Reviewer 1 Report
Comments and Suggestions for Authors
The review covers the rapidly developing topic of developing new classes of non-fullerene acceptors for organic solar cells, namely dimerized small-molecule acceptors. The review is well structured, the information is presented clearly and accessibly.
Among the minor comments I would like to note the following.
- Figures 1 and 2 are very small, it is impossible to see the structure of the molecules without zooming in. I understand that the structures are very voluminous, but perhaps the authors will find a way to make them more readable.
- In Figure 1, there is a missing a bond between two parts of a molecule in DIBP3F-S/Se.
- It seems that the molecules QM1 and QM1T are the same, and the molecules QM and 2BTP-2F-T differ only in the length of the alkyl substituents. Perhaps it will be possible to make this scheme more compact.
- In the π-spacer of GMA-SSeS, selenophene should be instead of thiophene.
- In Figure 3, it should be indicated what "n" is in the molecule DY-P2EH.
- Line 51: This is about 2017, and reference [18] refers to 2021.
- Line 288: This is about 2022, and reference [45] refers to 2023.
- Lines 92-105 refer to the substance DP-BTP shown in Figure 1. This should be indicated in the text.
- Line 269: A substance 2Qx-TT is mentioned that can be indicated in Figure 3 by n = 0.
- In references [32] and [33] it is necessary to write "A-π-A" instead of "A-π-a".
- In references [34] it is necessary to write "N-π-N" instead of "N-π-n".
- It might be worth mentioning Tables 1 and 2 in the text of the article.
Reviewer 2 Report
Comments and Suggestions for Authors
In this manuscript, the authors investigated two types of dimerized small-molecular acceptors, the conjugatively linked and the non-conjugatively linked, for organic solar cells. The article is very well-structured, the data are relevant, the references are appropriate, and the use of the English language is correct. Nonetheless, I would like to take this opportunity to offer a few observations regarding the manuscript, which I will outline below in the form of comments.
Comment 1.
In my opinion, the review is somewhat brief compared to other manuscripts of this type, and the number of references is insufficient. In the following comments, I will try to offer some suggestions to the authors on how they could expand and improve their work to provide readers with a more comprehensive understanding of the topic.
Comment 1.
The introduction should be supplemented with basic information, such as what organic solar cells are, what the donor-acceptor active layer is, and what materials have been used as acceptors before small molecules (e.g., fullerenes, non-fullerene acceptors, etc.). Here are some references I suggest:
https://doi.org/10.1557/JMR.2004.0252
https://doi.org/10.1002/anie.200702506
https://doi.org/10.1016/j.tsf.2019.137780
https://doi.org/10.1016/j.mtchem.2024.102290
Comment 2.
It is also worth mentioning thin-film deposition techniques, as they significantly contribute to the morphology of the active layer and, consequently, its efficiency (e.g., spin-coating, self-assembly, etc.). Here are some recommended references:
https://doi.org/10.3390/coatings12081115
https://doi.org/10.1002/adma.201204075
https://doi.org/10.1039/C8CS00376A
Comment 3.
I have found many more references from 2025 on dimerized small-molecular acceptors, and these should definitely be added to the manuscript.
Here are some recommended references:
https://doi.org/10.1016/j.cej.2025.160416
https://doi.org/10.1002/marc.202400687
https://doi.org/10.1016/j.mser.2024.100922
Round 2
Reviewer 2 Report
Comments and Suggestions for Authors
The authors have implemented the modifications I suggested, so I find the manuscript suitable for publication.